# Radial Access for Coronary Angiography Carries Fewer Complications Compared with Femoral Access: A Meta-Analysis of Randomized Controlled Trials

**DOI:** 10.3390/jcm10102163

**Published:** 2021-05-17

**Authors:** Gani Bajraktari, Zarife Rexhaj, Shpend Elezi, Fjolla Zhubi-Bakija, Artan Bajraktari, Ibadete Bytyçi, Arlind Batalli, Michael Y. Henein

**Affiliations:** 1Department of Public Health and Clinical Medicine, Umeå University, 90187 Umeå, Sweden; artanbajraktari@hotmail.com (A.B.); ibytyci@hotmail.com (I.B.); michael.henein@umu.se (M.Y.H.); 2Clinic of Cardiology, University Clinical Centre of Kosova, 10000 Prishtina, Kosovo; dr.zariferexhaj@gmail.com (Z.R.); selezi@hotmail.com (S.E.); fjolla.zhubi@gmail.com (F.Z.-B.); arlindbatalli@hotmail.com (A.B.); 3Medical Faculty, University of Prishtina “Hasan Prishtina”, 10000 Prishtina, Kosovo; 4UBT College, 10000 Prishtina, Kosovo

**Keywords:** femoral, radial, coronarography, PCI, acute coronary syndrome, stable coronary artery disease

## Abstract

Background and Aim: In patients undergoing diagnostic coronary angiography (CA) and percutaneous coronary interventions (PCI), the benefits associated with radial access compared with the femoral access approach remain controversial. The aim of this meta-analysis was to compare the short-term evidence-based clinical outcome of the two approaches. Methods: The PubMed, Embase, Cochrane Central Register of Controlled Trials, and ClinicalTrials.gov databases were searched for randomized controlled trials (RCTs) comparing radial versus femoral access for CA and PCI. We identified 34 RCTs with 29,352 patients who underwent CA and/or PCI and compared 14,819 patients randomized for radial access with 14,533 who underwent procedures using femoral access. The follow-up period for clinical outcome was 30 days in all studies. Data were pooled by meta-analysis using a fixed-effect or a random-effect model, as appropriate. Risk ratios (RRs) were used for efficacy and safety outcomes.Results: Compared with femoral access, the radial access was associated with significantly lower risk for all-cause mortality (RR: 0.74; 95% confidence interval (CI): 0.61 to 0.88; *p* = 0.001), major bleeding (RR: 0.53; 95% CI:0.43 to 0.65; *p* ˂ 0.00001), major adverse cardiovascular events (MACE)(RR: 0.82; 95% CI: 0.74 to 0.91; *p* = 0.0002), and major vascular complications (RR: 0.37; 95% CI: 0.29 to 0.48; *p* ˂ 0.00001). These results were consistent irrespective of the clinical presentation of ACS or STEMI. Conclusions: Radial access in patients undergoing CA with or without PCI is associated with lower mortality, MACE, major bleeding and vascular complications, irrespective of clinical presentation, ACS or STEMI, compared with femoral access.

## 1. Introduction

Patients with coronary artery disease (CAD) typically present with chest pain or shortness of breath. In patients with stable or unstable CAD, coronary angiography (CA), as the gold standard for detection and assessment of coronary artery stenoses, is performed, according to current clinical guidelines [1]. Revascularization therapy is indicated in patients with acute coronary syndrome (ACS) and in those with confirmed significant coronary stenosis not responding to optimal medical therapy or demonstrating marked limitation of physical activity [1]. Percutaneous coronary intervention (PCI), as an alternative to coronary artery bypass graft surgery, was introduced in the 1990s and is currently performed as a revascularization tool in the majority of patients with CAD [2]. The traditional approach for CA and PCI has been through the femoral artery, owing to its large caliber, which provides easy access [3]. Bleeding is the most common complication of PCI and is associated with poor clinical outcomes [4,5]. However, since 1989, the trans-radial approach has been attempted as an alternative to femoral access [6] and has resulted in less access-site bleeding due to the easily compressible radial artery; the superficial anatomy of the radial artery also encourages early patient discharge after procedures [7]. However, the radial approach for diagnostic CA and PCI requires a longer learning curve and higher procedure volumes in order to achieve adequate and safe skills. Over the last decades, several published randomized clinical trials (RCTs) assessed the value of the radial compared with the femoral approach in patients undergoing diagnostic CA and PCI with respect to residual ischemic, bleeding, and combined outcomes. The results of these RCTs remain controversial.

Therefore, in this meta-analysis, we aimed to provide a comprehensive and quantitative assessment of the available evidence from RCTs in comparing the clinical outcome of the radial and femoral approach to CA for diagnostic and interventional objectives.

## 2. Methods

Following the 2009 guidelines for systematic reviews and meta-analysis, we used PRISMA [8]. Based on the study design (meta-analysis), there was no need to request Institutional Review Board (IRB) approval or patient informed consent.

### 2.1. Search Strategy

We systematically searched PubMed-Medline, EMBASE, Scopus, Google Scholar, the Cochrane Central Registry of Controlled Trials and ClinicalTrial.gov, up to June 2020, using the following key words: (‘femoral’ OR ‘transfemoral’) AND (‘radial’ OR ‘transradial’) AND (‘percutaneous coronary intervention’ OR ‘PCI’ OR ‘coronarography’ OR ‘coronary angiography’) AND (‘randomized controlled trial’ OR ‘RCT’). Additional searches for potential trials included the references of review articles on that subject and the abstracts from selected congresses: scientific sessions of the European Society of Cardiology (ESC), the American Heart Association (AHA), the American College of Cardiology (ACC) and the European Society of Atherosclerosis (EAS). The literature search was limited to articles published in English. Two reviewers (GB and FZB) independently evaluated each article separately. No filters were applied. The remaining articles were obtained in fulltext and assessed again by the same two researchers, who evaluated each article independently and carried out data extraction and quality assessment. Disagreements were resolved by discussion with a third party (MYH).

### 2.2. Eligibility Criteria

Studies eligible for inclusion were those fulfilling the following criteria: (1) RCTs comparing the clinical outcome of the radial and femoral approach to CA for diagnostic and interventional objectives; (2) minimum follow-up of in-hospital stay; and (3) full-text studies published in peer-reviewed journals in English.Observational and unpublished studies were not included in the meta-analysis.

### 2.3. Data Extraction

Eligible studies were reviewed, and the following data were abstracted: (1) first author’s name; (2) year of publication; (3) name of clinical trial; (4) country where the study was performed; (5) number of centers; (6) study design; (7) number of patients in the two groups of unprotected LMCA revascularization; (8) follow-up duration and (9) clinical outcome data and number of events in both groups.

### 2.4. Clinical Outcomes and Definitions

The clinical outcomes of interest were evaluated at the longest available follow-up time(up to 30 days). There were 2 primary endpoints: all-cause mortality and major bleeding. Secondary efficacy and safety outcomes were myocardial infarction, stroke, the composite of major adverse cardiovascular events (MACE), and major vascular complications. The definition of MACE included the composite of death, stroke and myocardial infarction. Major bleeding was defined according to the scales used in each study [9,10], whereas major vascular complications were adjudicated according to the study definition or as hematoma >5 cm or pseudoaneurysm, if not reported in each study.

### 2.5. Quality Assessment

Risk of bias assessment in the included studies was evaluated by the same investigators for each study and was performed systematically using the Cochrane quality assessment tool for RCTs [11]. The Cochrane tool has 7 criteria for quality assessment: random sequence generation (selection bias), allocation sequence concealment (selection bias), blinding of participants and personnel (performance bias), blinding of outcome assessment (detection bias), incomplete outcome data (attrition bias), selective outcome reporting (reporting bias) and other potential sources of bias. The risk of bias in each study was classified as “low”, “high” or “unclear”.

### 2.6. Statistical Analysis

We performed the pooled analyses of clinical outcomes and treatment effects using the Cochrane Collaborative software, RevMan 5.3.5 (the Nordic Cochrane Center, the Cochrane Collaboration, 2014, Copenhagen, Denmark) [12]. A two-tailed *p* value < 0.05 was considered as significant. The baseline characteristics are reported as median and range. Mean and standard deviation (SD) values were estimated using the method described by Hozo et al. [13]. Analysis is presented in forest plots. Meta-analyses were performed with a fixed-effect model and random effect model, based on the encountered heterogeneity. Heterogeneity between studies was assessed using the Cochrane Q test and I^2^ index. As a guide, I^2^ < 25% indicated low, 25–50% moderate, and >50% high heterogeneity [14]. Publication bias was assessed using visual inspection of funnel plots and Egger’s test.

## 3. Results

### 3.1. Search Results and Trial Flow

Of the 334 articles identified in the initial search, 100 studies were screened as potentially relevant, but following critical scrutiny, only 34 RCTs [15,16,17,18,19,20,21,22,23,24,25,26,27,28,29,30,31,32,33,34,35,36,37,38,39,40,41,42,43,44,45,46,47,48] were considered appropriate and were included in this meta-analysis (Figure 1). The main characteristics of the included studies are reported in Appendix A.

### 3.2. Characteristics of Included Patients

Of the 29,352 patients eligible for analysis, 14,819 patients were assigned to the radial approach and 14,533 were assigned to the femoral approach. Random effect risk ratios were used for efficacy and safety outcomes. The mean age of patients was 59 years.

## 4. Outcomes of Patients in the Whole Group

### 4.1. Primary Clinical Outcomes

#### 4.1.1. All-Cause Mortality

All-cause mortality was reported in 33/34 included trials. All-cause mortality occurred in 191 patients (1.3%) assigned to the radial approach and in 262 patients (1.8%) assigned to the femoral approach at the latest follow-up. All-cause mortality was lower in patients assigned to the radial approach compared to those assigned to the femoral approach (RR: 0.74; 95% CI: 0.61 to 0.88; *p* = 0.001, Figure 2A). There was no evidence for heterogeneity across the RCTs (I^2^ = 0%).

#### 4.1.2. Major Bleeding

Major bleeding was reported in 29/34 included RCTs, having occurred in 607 patients (4.1%) assigned to the radial approach and in 1088 patients (8%) assigned to the femoral approach. Major bleeding was less frequent in patients assigned to the radial approach compared to those assigned to the femoral approach (RR: 0.53; 95% CI: 0.43 to 0.65; *p* ˂ 0.00001, Figure 2B). The heterogeneity was moderate (I^2^ = 28%).

### 4.2. Secondary Clinical Outcomes

#### 4.2.1. MACE

MACE was reported in all 34 included RCTs. MACE occurred in 605 patients (5%) assigned to the radial approach and in 745 patients (6.2%) assigned to the femoral approach at the latest follow-up. MACE werefewer in patients assigned to the radial approach compared to those assigned to the femoral approach (RR: 0.82; 95% CI: 0.74 to 0.91; *p* = 0.0002, Figure 3A). There was no evidence for heterogeneity across the RCTs (I^2^ = 0%).

#### 4.2.2. Major Vascular Complications

Major vascular complications were reported in 27/34 RCTs, having occurred in 148 patients (1.1%) assigned to the radial approach and in 399 patients (2.9%) assigned to the femoral approach at the latest follow-up. Major vascular complications were fewer in patients assigned to the radial approach compared to those assigned to the femoral approach (RR: 0.37; 95% CI: 0.29 to 0.48; *p* ˂ 0.00001, Figure 3B). The heterogeneity was moderate (I^2^ = 21%).

#### 4.2.3. Myocardial Infarction

Myocardial infarction was reported in 29/34 RCTs, occurring in 422 patients (3.1%) assigned to the radial approach and in 462 patients (3.3%) assigned to the femoral approach at the latest follow-up. Myocardial infarction was not different between patients assigned to the radial approach compared to those assigned to the femoral approach (RR: 0.92; 95% CI: 0.81 to 1.05; *p* = 0.20, Figure 3C). There was no evidence for heterogeneity across the RCTs (I^2^ = 0%).

#### 4.2.4. Stroke

Stroke was reported in 22/34 RCTs, having occurred in 62 patients (0.5%) assigned to the radial approach and in 58 patients (0.45%) assigned to the femoral approach at the latest follow-up. Stroke was not different between patients assigned to the radial approach compared to those assigned to femoral approach (RR: 1.10; 95% CI: 0.77 to 1.57; *p* = 0.60, Figure 3D). There was no evidence for heterogeneity across the RCTs (I^2^ = 0%).

## 5. Results

### 5.1. Outcomes of Patients with Acute Coronary Syndrome (ACS)

From the 34 RCTs included in this meta-analysis, 18 RCTs included patients with ACS.

#### 5.1.1. Primary Clinical Outcomes

##### All-Cause Mortality

All-cause mortality was reported in all 18 trials. All-cause mortality occurred in 179 patients (1.7%) assigned to the radial approach and in 245 patients (2.3%) assigned to the femoral approach at the latest follow-up. All-cause mortality was lower in patients assigned to the radial approach compared to those assigned to the femoral approach (RR: 0.73; 95% CI: 0.61 to 0.89; *p* = 0.001, Appendix A). There was no evidence for heterogeneity across the RCTs (I^2^ = 0%).

##### Major Bleeding

Major bleeding was reported in 17/18 RCTs, having occurred in 573 patients (5.1%) assigned to the radial approach and in 984 patients (9.3%) assigned to the femoral approach. Major bleeding occurred lessoften in patients assigned to the radial approach compared to those assigned to the femoral approach (RR: 0.63; 95% CI: 0.57 to 0.70; *p* ˂ 0.00001, Appendix A). The heterogeneity was moderate (I^2^ = 37%).

#### 5.1.2. Secondary Clinical Outcomes

##### MACE

MACE was reported in all 18 included RCTs, in which 542 patients (6.7%) were assigned to the radial approach and 657 patients (8%)were assigned to the femoral approach at the latest follow-up. MACE were fewer in patients assigned to the radial approach compared to those assigned to the femoral approach (RR: 0.83; 95% CI: 0.74 to 0.92; *p* =0.0007, Appendix A). There was no evidence for heterogeneity across the RCTs (I^2^ = 0%).

##### Major Vascular Complications

Major vascular complications were reported in 16/18 RCTs, having occurred in 132 patients (1.3%) assigned to the radial approach and 308 patients (2.9%) assigned to the femoral approach at the latest follow-up. Major vascular complications were less in patients assigned to the radial approach compared to those assigned to the femoral approach (RR: 0.44; 95% CI: 0.36 to 0.53; *p* ˂ 0.00001, Appendix A). The heterogeneity was moderate (I^2^ = 27%).

##### Myocardial Infarction

Myocardial infarction was reported in 17/18 RCTs, having occurred in 392 patients (3.7%) assigned to the radial approach and in 432 patients (4.1%) assigned to the femoral approach at the latest follow-up. Myocardial infarction was not different between patients assigned to the radial compared to the femoral approach (RR: 0.91; 95% CI: 0.80 to 1.04; *p* = 0.17, Appendix A). There was no evidence for heterogeneity across the RCTs (I^2^ = 0%).

##### Stroke

Stroke was reported in 11/18 included RCTs. Stroke occurred in 62 patients (0.6%) assigned to the radial approach and in 51 patients (0.5%) assigned to the femoral approach at the latest follow-up. Stroke was not different between patients assigned to the radial approach compared to those assigned to the femoral approach (RR: 1.22; 95% CI: 0.85 to 1.76; *p* = 0.29, Appendix A). There was no evidence for heterogeneity across the RCTs (I^2^ = 0%).

### 5.2. Outcomes of Patients with STEMI

Of the 34 RCTs included in this meta-analysis, 14 RCTs had patients with STEMI.

#### 5.2.1. Primary Clinical Outcomes

##### All-Cause Mortality

All-cause mortality was reported in all 14 trials, having occurred in 124 patients (2.3%) assigned to the radial approach and in 181 patients (3.2%) assigned to the femoral approach at the latest follow-up. All-cause mortality was lower in patients assigned to the radial approach compared to those assigned to the femoral approach (RR: 0.69; 95% CI: 0.56 to 0.87; *p* = 0.001, Appendix A). There was no evidence for heterogeneity across the RCTs (I^2^ = 0%).

##### Major Bleeding

Major bleeding was reported in all 14 included RCTs, with 46 patients (1.3%) assigned to radial approach and 95 patients (2.7%) assigned to femoral approach at the latest follow-up. Major bleeding was lower in patients assigned to the radial approach compared to those assigned to the femoral approach (RR: 0.57; 95% CI: 0.51 to 0.64; *p* ˂ 0.00001, Appendix A). There was no evidence for heterogeneity across the RCTs (I^2^ = 0%).

#### 5.2.2. Secondary Clinical Outcomes

##### MACE

MACE was reported in all 14 included RCTs, having occurred in 265 patients (4.8%) assigned to the radial approach and in 319 patients (5.7%) assigned to the femoral approach at the latest follow-up. MACE was less in patients assigned to the radial approach compared to those assigned to the femoral approach (RR: 0.84; 95% CI: 0.72 to 0.98; *p* = 0.03, Appendix A). There was no evidence for heterogeneity across the RCTs (I^2^ = 0%).

##### Major Vascular Complications

Major vascular complications were reported in 13/14 RCTs, with 71 patients (2.1%) assigned to the radial approach and 148 patients (4.2%) assigned to the femoral approach at the latest follow-up. Major vascular complications were fewer in patients assigned to the radial approach compared to those assigned to the femoral approach (RR: 0.48; 95% CI: 0.37 to 0.62; *p* ˂ 0.00001, Appendix A). The heterogeneity was moderate (I^2^ = 31%).

##### Myocardial Infarction

Myocardial infarction was reported in 13/14 RCTs, having occurred in 33 patients (1.3%) assigned to the radial approach and in 35 patients (1.4%) assigned to the femoral approach at the latest follow-up. Myocardial infarction was not different between patients assigned to the radial compared to the femoral approach (RR: 0.92; 95% CI: 0.80 to 1.04; *p* = 0.19, Appendix A). There was no evidence for heterogeneity across the RCTs (I^2^ = 0%).

##### Stroke

Stroke was reported in 10/14 included RCTs, with 19 patients (0.8%) assigned to the radial approach and 14 patients (0.6%) assigned to the femoral approach at the latest follow-up. Stroke was not different between patients assigned to the radial compared to the femoral approach (RR: 1.25; 95% CI: 0.86 to 1.80; *p* = 0.24, Appendix A). There was no evidence for heterogeneity across the RCTs (I^2^ = 0%).

#### 5.2.3. Risk of Bias Assessment

The assessment of risk of bias and applicability concerns based on the Quality Assessment of Diagnostic Accuracy Studies questionnaire (QUADAS-2) was used for our study questions [8]. All of the criteria domains for risk of bias and applicability were analyzed. The risk of bias was assessed as ‘‘low risk,’’ ‘‘high risk,’’ or ‘‘unclear risk’’.Most studies had high a quality (high or moderate level) and clearly defined objectives and main outcomes (Appendix A). All domains had low risk of bias (<20%) and no evidence for publication bias based on the Egger’s test.

## 6. Discussion

### 6.1. Findings

This systematic review and meta-analysis consisted of 34 RCTs with 29,352 patients undergoing diagnostic CA for suspected significant stenotic CAD, including those who underwent PCI, if it was indicated. Included patients were randomized to either the radial or femoral approach. The main findings of this meta-analysis are as follows. (1) The use of radial access compared with femoral access was associated with a significant 26% relative risk reduction in all-cause mortality and 47% relative risk reduction in major bleeding in all patients undergoing CA. The risk reduction was also demonstrated irrespective of the clinical presentation of ACS (27% and 37%, respectively) or STEMI (31% and 43%, respectively). (2) The use of radial access was associated with 18% fewer MACE and 63% fewermajor vascular complications. Again, the radial approach was associated with lower risk of MACE, irrespective of clinical presentation of ACS (17% and 56%, respectively) or STEMI (16% and 52%, respectively) (Figure 4). (3) There was no significant difference in myocardial infarction and stroke between patients assigned to radial approachwith respect to those assigned to the femoral approach in the two subgroups with ACS or STEMI (Figure 4).

### 6.2. Data Interpretation

Coronary angiography and PCI procedures are used in the diagnosis and management of CAD [49]. The traditional approach for CA and PCI, from their introduction, has been through the femoral artery, based on the easy access it offers due to the large caliber [3]. The transfemoral approach for CA and PCI gained widespread acceptance by operators because of the following advantages: long history of use, easy technicality, and the ability of clinicians to use larger catheters and equipment for various interventions [50]. However, the femoral catheterization approach has some disadvantages, including the need for patients’ prolonged bed rest, which could be associated with back pain, urinary retention, and neuropathy, particularly in the elderly [6,51,52],as well as prolonged arterial compression related complications, e.g., peripheral ischemia. These limitations promptedcardiologists to explore attempting the trans-radial approach as an alternative, with the first procedure performed in 1989 by Campeau [6], which demonstrated excellent results. Since then, trans-radial catheterization has been an attractive option for both operators and patients, despite the need for prolonged training for optimum skill development. Furthermore, it has also been reported that the trans-radial approach to CA and PCI carries significantly fewervascular complications, including pseudoaneurysms, arteriovenous fistulas, bleeding, and retroperitoneal hematomas, compared with the transfemoral approach [28].

Such advantages could be explained on the basis of reduced limb-threatening ischemia, less need for lying flat (e.g., due to back pain, obesity, or congestive heart failure), and earlier patient discharge. The reduced limb-threatening ischemia could be explained on the basis of a lower likelihood of radial atherosclerosis disease, which is disease known to affect femoral arteries [53]. The technique also does not cross the descending and thoracic aorta, which are known for their involvement in the process of atherosclerosis; hence, this process does not involve the potential thromboembolic complications of the femoral approach [53]. On the other hand, the main limitations of the trans-radial approach are the relatively smaller caliber [54], potential technical challenges in some patients, potential vessel spasm [54], longer procedure time requiring higher radiation [55] and a steep learning curve [56,57]. Despite these limitations, data from the last decades support the use of radial access as the default approach for CA and PCI in the whole spectrum of patients with suspected CAD who are undergoing invasive diagnosis and PCI.

The results of our meta-analysis explain objectively the strong support of the change from the traditional “femoral first” paradigm to the “radial first” approach. In particular, the significantly lower levels of all-cause mortality and major bleeding with the radial approach compared to the femoral approach addressesthe controversial findings between the two approaches ([42,43,48,58] and [43,48], respectively) as well as other clinical outcomes [42,43,48,58]. Our findings also support the use of the trans-radial approach even in patients with ACS and STEMI, in whom serious complications proved to be significantly reduced compared to the femoral approach, despite the results of the recently published SAFARI-STEMI RCT [47], which showed comparable outcomes in both approaches.

### 6.3. Clinical Implications

Our meta-analysis supports the current European Society of Cardiology recommendations that trans-radial coronary artery catheterization is the default procedure unless other technical or anatomical limitations require the use of the alternative femoral approach. The associated lower clinical risk, including MACE and mortality, further supports the trans-radial coronary catheterization approach.

### 6.4. Limitations

As is the case with all meta-analyses, we relied on the published data from randomized clinical trials, so we did not have any hand in the accuracy of the data collection; however, there is no reason to doubt the high scientific level of these trials. If there were weaknesses in one of the RCTs, this did not seem to have any significant impact on the overall results of the meta-analysis. The over 14,000 patients randomized to each investigation arm—transfemoral and trans-radial—strengthen the analysis results and demonstrate consistently strong findings. Comparing the overall meta-analysis findings with some individual RCT results could show differences; however, such analyses should be considered as strong evidence over and above individual studies, irrespective of their size. Another conflicting opinion is the relationship between bleeding events and the use of different antithrombotic/anticoagulation protocols [58,59,60,61,62] in patients undergoing radial versus femoral approaches. These protocols should not affect the results of our meta-analysis, as all included studies were RCTs.

### 6.5. Conclusions

The results of this meta-analysis support the superior clinical safety of using trans-radial coronary artery catheterization, with or without intervention, over and above the traditional transfemoral approach.

## Figures and Tables

**Figure 1 jcm-10-02163-f001:**
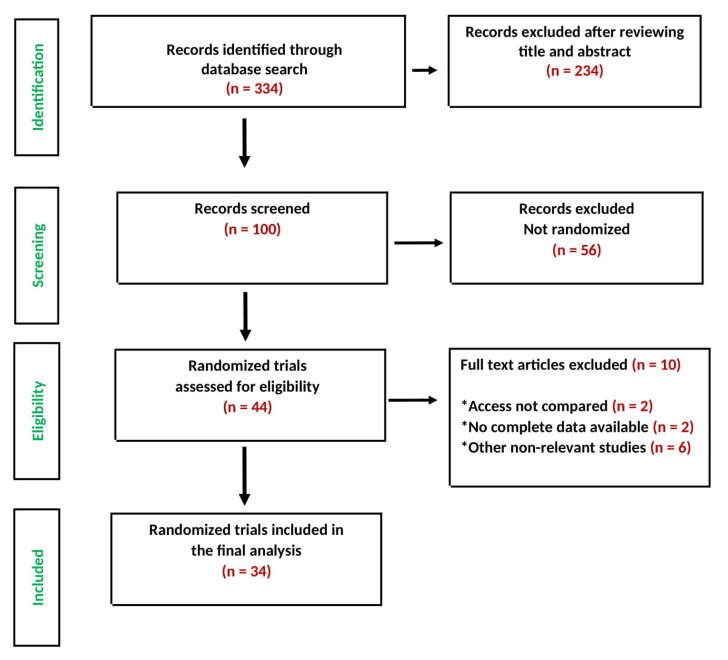
PRISMA study selection flow chart.

**Figure 2 jcm-10-02163-f002:**
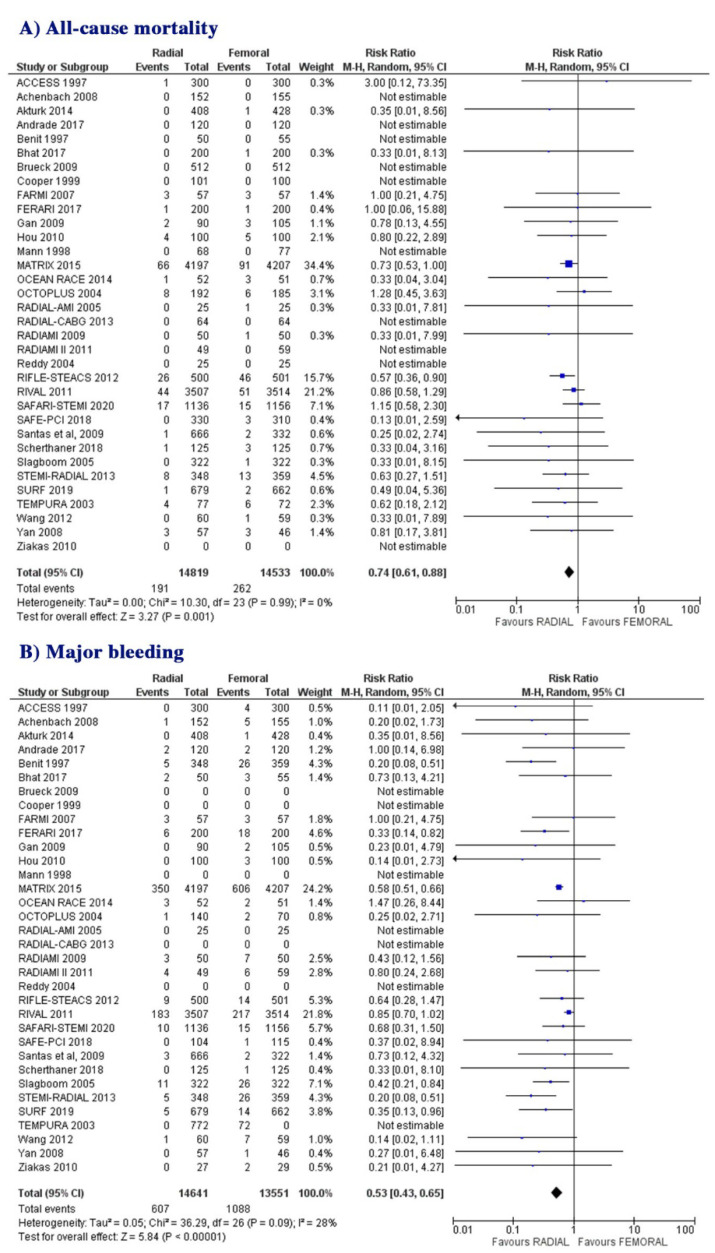
Risk of all-cause mortality (**A**) and major bleeding (**B**) at follow-up: radial vs. femoral, in whole group of patients.

**Figure 3 jcm-10-02163-f003:**
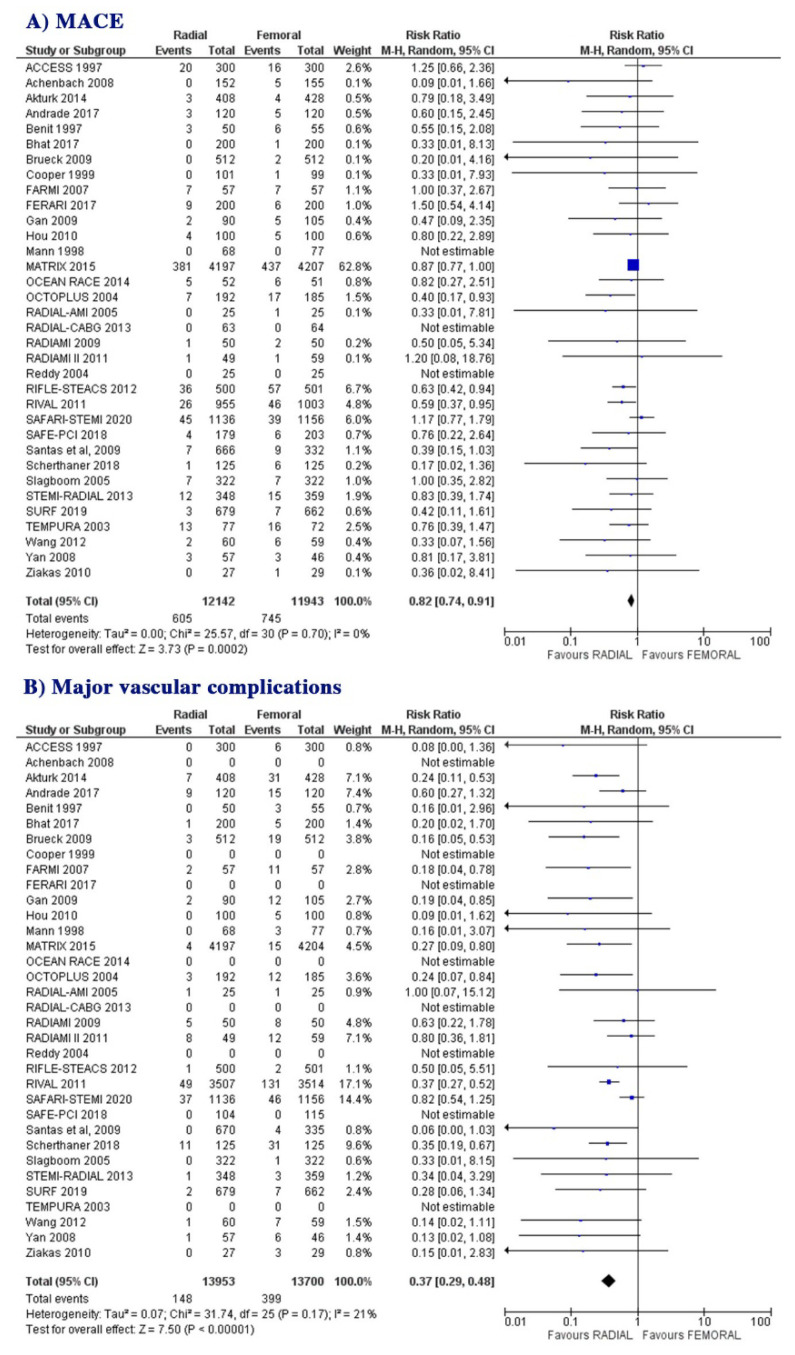
Risk of MACE (**A**) and major vascular complications (**B**) at follow-up: radial vs. femoral, in whole group of patients.Risk of myocardial infarction (**C**) and stroke (**D**) at follow-up: radial vs. Femoral, in whole group of patients.

**Figure 4 jcm-10-02163-f004:**
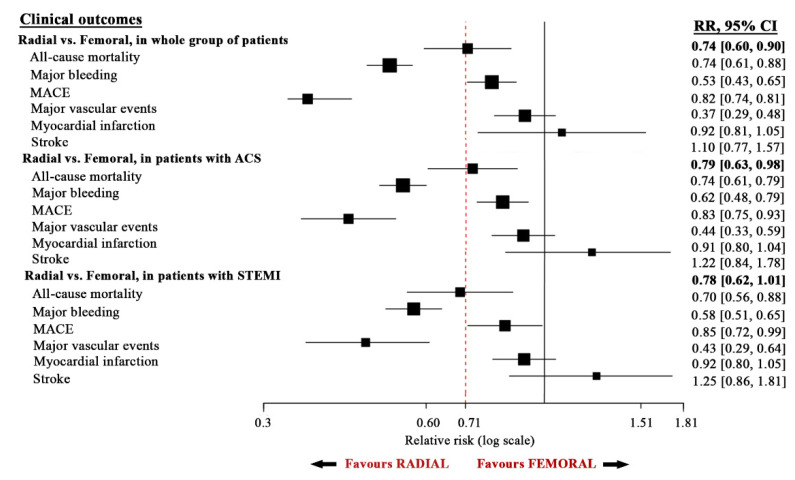
Summary of outcome in all study groups.

## Data Availability

Not applicable.

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
