# Peer review of "Radial Access for Coronary Angiography Carries Fewer Complications Compared with Femoral Access: A Meta-Analysis of Randomized Controlled Trials"

_jcm, 2021, doi:10.3390/jcm10102163_

Round 1

Reviewer 1 Report

This paper is a meta-analysis that investigated the complication rate of radial access versus femoral access during coronary angiography. The study has revealed some interesting findings. However, after reviewing the manuscript, there are several issues that need to be addressed prior to publication:

  • This study is a meta-analysis. The study did not specify whether the cases for femoral access were preferentially chosen over radial access or whether femoral access was performed because of inability to access the radial artery. In the latter case, this could skew the data as patients who require femoral access could be in more serious conditions to begin with than patients who can be accessed via radial artery.
  • In this day and age, all vascular access should be guided by ultrasound instead of by the landmark approach to avoid possible vascular injury. Was the difference in outcomes between US versus landmark guidance considered when analyzing the data? If not, femoral access without US guidance would certainly bear higher injury risk and bleeding under anticoagulation.
  • In one of the analyses, the authors have associated all-cause mortality with invasive arterial access, but this association may not have any cause-effect relationship as, again, patients needing femoral access may have more serious conditions to begin with.
  • Did authors consider whether radial artery access is a viable option in all ACS and STEMI conditions?

Author Response

This paper is a meta-analysis that investigated the complication rate of radial access versus femoral access during coronary angiography. The study has revealed some interesting findings. However, after reviewing the manuscript, there are several issues that need to be addressed prior to publication:

This study is a meta-analysis. The study did not specify whether the cases for femoral access were preferentially chosen over radial access or whether femoral access was performed because of inability to access the radial artery. In the latter case, this could skew the data as patients who require femoral access could be in more serious conditions to begin with than patients who can be accessed via radial artery.

Response: We thank you for the question. We included in our meta-analysis only Randomized Controlled Trials, so all patients included in this meta-analysis were randomized. 

In this day and age, all vascular access should be guided by ultrasound instead of by the landmark approach to avoid possible vascular injury. Was the difference in outcomes between US versus landmark guidance considered when analyzing the data? If not, femoral access without US guidance would certainly bear higher injury risk and bleeding under anticoagulation.

  • Response: We thank you for the question. The Randomized Controlled Trials that we included in this meta-analysis are done in different times, and the objective of these trials was not to compare US guidance vs. non-US guidance. This could be a field for a future study

In one of the analyses, the authors have associated all-cause mortality with invasive arterial access, but this association may not have any cause-effect relationship as, again, patients needing femoral access may have more serious conditions to begin with.

  • Response: We thank you for the question. We agree on your argument but non of the included trials mentioned worse vascular/cardiac condition as a reason for femoral access. In addition, all included patients in this meta-analysis were randomized, as we included only RCTs. 

Did authors consider whether radial artery access is a viable option in all ACS and STEMI conditions?

  • Response: We thank you for the question. Yes, we first assessed all included patients, and secondly, we assessed patients with ACS and those with STEMI. These results are shown in the Supplementary Figures. 

Reviewer 2 Report

Overall well done meta analysis of a controversial topic; large number of patients were included and showed a small (<1%) absolute difference in mortality between radial vs femoral.  

AHA / ACC guidelines do not currently support the following claim; consider rewording this statement "Our meta-analysis supports the current recommendations that trans-radial coronary artery catheterization is the default procedure unless other technical or anatomical limitations require the use of the alternative femoral approach"

the largest randomized trials (Safari, Rival, Matrix) do not show reductions in MACE/mortality ... how do we reconcile this? 

Could you include a table of the studies and how many patients from each are included in your analysis ...I was unable to find this...

Would be nice to compared data in patients receiving only PCI or diagnostic cath without PCI? Can we compared data amonst patients receiving bivalirudin or gp2b3a ?

Author Response

Overall well done meta analysis of a controversial topic; large number of patients were included and showed a small (<1%) absolute difference in mortality between radial vs femoral.  

  • Response: We thank you for your statement. We agree. 

AHA / ACC guidelines do not currently support the following claim; consider rewording this statement "Our meta-analysis supports the current recommendations that trans-radial coronary artery catheterization is the default procedure unless other technical or anatomical limitations require the use of the alternative femoral approach"

  • Response: We thank you for the question. We have added to these sentence “…European Society of Cardiology…”. (Page 15).

The largest randomized trials (Safari, Rival, Matrix) do not show reductions in MACE/mortality ... how do we reconcile this? 

  • Response: We thank you for the question. We agree completely with your statement. This was also our main objective of this meta-analysis in order to see the results after pooling all available data. 

Could you include a table of the studies and how many patients from each are included in your analysis ...I was unable to find this...

  • Response: We thank you for the question. We have included the table as supplementary file. 

Would be nice to compared data in patients receiving only PCI or diagnostic cath without PCI? Can we compared data among patients receiving bivalirudin or gp2b3a ?

  • Response: We thank you for the question. The problem is that most studies did not separate data of patients receiving only diagnostic coronary angiography. However, inpatients with ACS and STEMI, all patients received PCI.   

Relying on the included RCTs, it seems that most patients did not receive Bivalirudin, since most of studies even did not mention it, as the objective of these studies was only to compare radial vs. femoral approach in randomized patients.

Round 2

Reviewer 2 Report

Results differ from several large RCTs, please elaborate on why you results are so different and would address or update some of the limitations of your paper (diagnostic and PCI etc)

Author Response

We thank you for your suggestion. We added in the limitation section the sentence: “However, comparing the overall meta-analysis findings with individual RCT results could show differences, hence the conventional approach in considering such analyses as strong evidence over and above individual studies, irrespective of its size.”.